# Cohort Profile: Effectiveness of a 12-Month Patient-Centred Medical Home Model Versus Standard Care for Chronic Disease Management among Primary Care Patients in Sydney, Australia

**DOI:** 10.3390/ijerph17062164

**Published:** 2020-03-24

**Authors:** James Rufus John, Amanda Jones, A. Munro Neville, Shima Ghassempour, Federico Girosi, W. Kathy Tannous

**Affiliations:** 1Translational Health Research Institute, Western Sydney University, New South Wales 2560, Australia; F.Girosi@westernsydney.edu.au (F.G.); k.tannous@westernsydney.edu.au (W.K.T.); 2Rozetta Institute, Level 4, 55 Harrington Street, Sydney, New South Wales 2000, Australia; 3Sonic Clinical Services, Level 21, 225 George Street, Sydney, New South Wales 2000, Australia; amanda.jones@scs.com.au; 4AusTrials, Queensland 4068, Australia; munro.neville@austrials.com.au; 5Research Implementation Science and eHealth Group, Faculty of Health Science, The University of Sydney, New South Wales 2006, Australia; shima.ghassempour@sydney.edu.au; 6School of Business, Western Sydney University, New South Wales 2150, Australia

**Keywords:** Patient-centred medical home, enhanced primary care, general practices, integrated care, chronic care model

## Abstract

Evidence suggests that patient-centred medical home (PCMH) is more effective than standard general practitioner care in improving patient outcomes in primary care. This paper reports on the design, early implementation experiences, and early findings of the 12-month PCMH model called ‘WellNet’ delivered across six primary care practices in Sydney, Australia. The WellNet study sample comprises 589 consented participants in the intervention group receiving enhanced primary care in the form of patient-tailored chronic disease management plan, improved self-management support, and regular monitoring by general practitioners (GPs) and trained clinical coordinators. The comparison group consisted of 7750 patients who were matched based on age, gender, type and number of chronic diseases who received standard GP care. Data collected include sociodemographic characteristics, clinical measures, and self-reported health assessments at baseline and 12 months. Early study findings show the mean age of the study participants was 70 years with nearly even gender distribution of males (49.7%) and females (50.3%). The most prevalent chronic diseases in descending order were circulatory system disorders (69.8%), diabetes (47.4%), musculoskeletal disorders (43.5%), respiratory diseases (28.7%), mental illness (18.8%), and cancer (13.6%). To our knowledge, the WellNet study is the first study in Australia to generate evidence on the feasibility of design, recruitment, and implementation of a comprehensive PCMH model. Lessons learned from WellNet study may inform other medical home models in Australian primary care settings.

## 1. Introduction

The growing burden of non-communicable diseases including cardiovascular diseases (CVD), diabetes, chronic obstructive pulmonary disease (COPD), mental illnesses, lung cancer, and musculoskeletal disorders is a major cause of disability and death [1,2]. In Australia, chronic diseases contributed towards 61% of the total disease (fatal and non-fatal) burden and 87% of all deaths in 2015 [3,4]. Furthermore, the burden of multiple chronic conditions (‘multimorbidity’) is a major public health issue with recent findings reporting that the prevalence of Australians with two or more and three or more conditions is 26% and 16%, respectively [5]. Studies show that patients with multimorbidity often experience poor health-rated quality of life (HR-QOL) [6], psychological distress [7], and increased mortality [8]. Multimorbidity is also associated with increased hospital admissions [9], health care expenditure [10], and inappropriate polypharmacy [11]. Furthermore, evidence suggests that the health burden of chronic diseases is projected to increase in the future, thereby challenging health systems worldwide to revisit strategies towards effective management and prevention [12,13].

Over the last few decades, advancements in public health policies and evidence-based medical treatments have contributed to increased life expectancy [14]. Consequently, the higher life span has resulted in a greater number of patients with comorbidities and, subsequently, a greater demand for health services [15,16]. Health care systems in high-income countries, including Australia, are primarily focused on the ‘single-disease framework’, where care delivery for the management of multimorbidity is often fragmented, lacking integration and continuity of care [17]. Conversely, studies show that coordinated and collaborative approaches in primary care with strong emphasis on self-management of chronic diseases are effective in managing complex multimorbidity [18,19]. Therefore, the rising demand for effective management of complex multimorbidity requires enhanced models of primary care for better patient and health service delivery outcomes [20].

Primary health care is the cornerstone of Australia’s health care system, providing continual, comprehensive and coordinated care that is targeted towards patients’ healthcare needs [21,22]. The country’s current primary health care system is built around a strong general practice foundation and it is estimated that 85 percent of Australians consulted a GP at least once annually [3]. A number of initiatives have recently been undertaken to integrate primary care for the management chronic diseases by government and primary care organisations, including the ‘Australian Better Health Initiative’, ‘National Primary Health Care Strategy Framework’, and ‘Australian Primary Care Collaboratives Program’ [23,24]. Though encouraging, primary care-based data and research on the effectiveness and feasibility of these initiatives providing a coordinated, multi-disciplinary team (MDT) care for long-term chronic disease management remains limited and not definitive [25,26].

The patient-centred medical home (PCMH) model has been lauded as providing the best model of primary care for patients with one or more chronic diseases. This is due to provision of continuous, comprehensive, and MDT care for collaborating services to meet patients’ health care needs [27]. Although definitions vary [28,29], the PCMH model typically includes a general practitioner (GP) as part of a MDT, working in conjunction with patients to provide coordinated and focussed care that promotes long-term patient engagement using a long-term chronic disease approach. There is a small but growing body of evidence, primarily from the United States, suggesting that various models of PCMH primary care are more effective than standard care in improving clinical outcomes in patients with one or more chronic diseases [30,31], increasing the quality of care delivered [32,33] and reducing hospital admissions [34,35]. However, in Australia, PCMH models have not been evaluated given the country’s health care setting and funding models. Therefore, the aim of this article is to report on the design, implementation, and early findings of the 12-month PCMH model called ‘WellNet’ delivered across six primary care practices in Sydney, Australia. Additionally, this article will also detail the overall aims and objectives of the WellNet study and the ongoing research activities conducted to evaluate the study outcomes.

## 2. Materials and Methods

### 2.1. Design of the WellNet Chronic Disease Management (CDM) Program

Sonic Clinical Services (SCS) developed the 12-month WellNet chronic disease management (‘WellNet’) program in 2016. This is closely aligned to the principles of the PCMH primary care model as it aims to deliver a coordinated and MDT model of care tailored directly to the needs of patients according to the level of risk and complexity of their chronic conditions. This includes health coaching, care navigation, education, self-management and regular review (Figure 1).

The WellNet program combined face-to-face and telephone consultations with care coordinators and GPs to provide optimal health outcomes through the delivery of patient-tailored healthcare interventions. The interventions are based on the key pillars of:Patient identification and enrolment;Patient centred integrated care;Outcomes-based program philosophy;Data analytics, risk assessment and patient stratification;Evidence-based interventions;Shared electronic health records; andPatient education and teaching self-management skills.

The WellNet program structured interventions are based on the best available models of care applicable to the primary care setting. The program is authored by Australian and international institutes, including the Royal Australian College of General Practitioners (RACGP), Diabetes Australia, Australian Lung Foundation, Therapeutic Guidelines Expert Groups and the National Heart Foundation [36,37,38,39,40,41,42,43,44,45,46,47,48,49,50,51]. As patients’ clinical presentations and needs are heterogeneous, the structured interventions are adapted to meet an individual patient’s needs within the IT platform, cdmNET, and with the guidance of the patients’ usual GP.

A key component of the program is the involvement of specialised chronic disease management (CDM) care coordinators at the medical centres. This was identified as key to successful integration of a CDM program by the literature [52,53]. Supporting the care coordinators were a wide range of services and structured interventions. These included online and print educational materials directed at encouraging behavioural change and a technology platform that facilitates the delivery of interventions across a broad healthcare delivery team.

In addition, patients were provided access to a user-friendly application called ‘GoShare’ which enables sharing of a range of health resources including video series, links to credible websites, apps, and tools tailored to patients’ information needs. This program is aimed to improve self-efficacy, self-management behaviours, and empower patients to play a more active role in their health care decisions [54].

The WellNet program consisted of 7 in-practice visits and 3 telephone contacts with the care coordinator (2 contacts linked with GP appointments on commencement of the program) and quarterly GP reviews (4 visits) making up to a total of 14 contacts. Flexibility with the number of contacts was also provided according to patient’s needs and availability. Therefore, patients’ clinical and self-reported outcomes were collected over 12 months plus or minus 3 months.

### 2.2. Specific and Broader Aims and Rationale of Study Design

This article aims to describe the design, implementation, and early findings of the 12-month PCMH model called ‘WellNet’ delivered across six primary care practices in Sydney, Australia.

The broader aim of the WellNet study is to evaluate the effectiveness of a PCMH model for improving clinical outcomes in primary care patients. The specific objectives of this study are to: (1) evaluate changes in clinical outcomes in study participants compared to patients receiving standard care between baseline and 12 months; (2) assess changes in participants’ self-reported health-related quality of life (HR-QoL) and level of activation; (3) determine changes in the risk of hospital admissions; (4) evaluate changes in disease-specific risk assessments; and (5) explore predictors of treatment uptake, response, and compliance.

A cohort study design with a comparison group (aim (1)) and a case-series study design (aims (1) to (5)) was used to evaluate WellNet’s ‘between group’ and ‘within group’ effectiveness. This was for the WellNet program delivered in six general practices in Northern Sydney, New South Wales, Australia. The rationale for using a synthetic comparison group was to have a sample that would be similar to the WellNet treatment cohort at baseline, which in turn would be used to test the hypothesis that WellNet group will have better clinical outcomes at 12 months compared to standard care.

### 2.3. Patient Characteristics

#### 2.3.1. Treatment Group—Enrolment Methods and Outcome

The WellNet study’s treatment group constituted 636 patients from six primary care practices in Northern Sydney who met the eligibility criteria to participate in the 12-month program and provided written consent to have their data shared for evaluation.

The recruitment period for the study was between December 2016 and October 2017. SCS developed and executed a computerised algorithm to identity from electronic medical records those patients who met the diagnosis and risk factor criteria shown in Figure 2. The WellNet risk algorithm categorised patients into four groups of complexity based on the number of chronic conditions and presence of risk factors. Patients in groups C and D (the more complex cases) were the target groups for the intervention, however eligibility was confirmed at initial assessment by Hospital Admission Risk Profile (HARP) score.

Patients were eligible if they satisfied either criteria:

Criteria 1: Patients aged 40 years or above and who had seen a GP at least three times in the last two years; had been diagnosed with one to three chronic diseases and had presented with one or more elevated clinical risk factors; and held a HARP score of greater than 10 (medium risk or greater) OR

Criteria 2: Patients aged 40 years or above and who had seen a GP at least three times in the last two years and had been diagnosed with four or more chronic diseases with or without one or more risk factors; and held a HARP score of greater than 10 (medium risk or greater).

Additionally, patients with low HARP score (<10) but with at least one or more chronic diseases and one or more consistently elevated risk factors were also included in the study through direct GP referrals as GPs deemed them as good candidates that could benefit from the WellNet care.

The care coordinators eliminated unsuitable patients, such as those living in nursing homes or with significant cognitive impairment, before presenting a list of potentially eligible patients to GPs for their review and selection. Potentially eligible patients were then contacted either through an invitation letter (n = 1431) or by GP referrals during routine visits (n = 359). A minimum of three follow-up phone calls were made by the care coordinator for each patient invited by letter who did not respond directly to the letter.

Out of the total 1790 patients contacted, 698 (38.9%) attended the initial assessment. From the initial assessment, 688 patients were found to be eligible for the program based on their HARP score. Of these, 52 declined to participate in the WellNet Program or the cohort study, resulting in 636 (92.4%) consenting eligible participants enrolled into the study.

#### 2.3.2. Comparison Group—Matching Iterations and Outcomes

For matching purposes, four general practices with similar geographical proximity as WellNet practices that did not provide PCMH care were chosen. In order to be concurrent with the enrolment period, patients who visited any one of the four general practices between December 2016 and October 2017 were identified (n = 20,478).

The Coarsened Exact Matching (CEM) was used to match treatment participants with the comparison group. CEM is a Monotonoic Imbalance Bounding (MIB) matching method that temporarily coarsens the data according to the researchers’ choice and then finds exact matches so that adjusting the imbalance on one variable has no effect on the maximum imbalance of any other [55]. Using ‘Coarsened Exact Matching’ analysis in R, five different matching iterations were conducted. Several variables including age (continuous and categorical), gender, chronic disease type (cardiovascular disease, respiratory disease, diabetes, musculoskeletal disease, mental illness, and cancer), number of chronic diseases, systolic blood pressure (continuous and categorical), and total cholesterol: HDL-C ratio were considered to determine the best possible matching outcome, and from that, the best model was chosen.

Subclass or strata groups without at least 1 treatment to 1 comparison patient were excluded. CEM in turn returns ‘weights’ to compensate for differential strata sizes to be used in the subsequent analyses [55]. This matching procedure has shown to effectively limit selection bias and variance between the groups, thereby minimising bias in the final model showing treatment effects [55].

Table 1 provides the matching iteration numbers by the variables and their outcomes. Matching iteration number 1 was chosen as the best model as it produced the highest number of matched patients in both treatment and comparison groups whilst closely matching the two groups based on age and gender, along with the same type and number of chronic diseases. Of the WellNet program group of 636 patients, 589 were matched to the comparison group (Figure 3).

### 2.4. Provider Characteristics

A total of 66 GPs were invited to participate in a Likert scale survey on their perspectives on (1) chronic disease management; (2) WellNet program; and (3) PCMH model of care. In addition, some sociodemographic characteristics on age, gender, place of graduation and years of general practice experience were also collected.

### 2.5. Follow-Up

There are two phases of follow-up in the WellNet program. Phase 1 involved follow-up at 12 months to evaluate changes in clinical outcomes, HARP, Patient Activation Measure (PAM), self-reported EuroQol 5 Dimensions 5 Levels (EQ-5D-5L), and other disease-specific risk assessments [49,56,57,58,59,60,61,62,63,64,65]. Phase 2 of the evaluation is planned as a two-year post intervention follow-up that will study changes in health services utilisation and medication prescription with the use of hospital-linked administrative datasets.

### 2.6. Data Collection

Data collected at various stages of baseline and follow-up from participants included sociodemographic information, private health insurance membership status, lifestyle risk factors, chronic disease diagnoses, clinical measures (as clinically relevant), and several validated health-related surveys that participants were asked to complete upon enrolment (Table 2). These surveys involved information regarding:(a)health-related quality of life,(b)level of health engagement and activation,(c)self-management of their health,(d)other disease-specific risk assessments.

### 2.7. Statistical Analyses

Data extraction on the WellNet program group and the comparison group were provided by SCS to the researchers. The data were provided in de-identified form and the patient’s cdmNET number was used to track patients over time.

#### 2.7.1. Analysis Conducted to Date

Descriptive statistics are presented as mean and standard deviation (SD) for continuous variables whereas frequency counts and percentages are used for reporting categorical variables. Independent samples t-test was conducted to determine significant differences between treatment and comparison group at baseline.

#### 2.7.2. Ongoing Analysis

To assess changes over time between baseline and 12 months, significant within-group mean differences (pre-test/post-test) from baseline to 12 months will be determined by using Paired samples t-test. In addition, between-group analyses (treatment and comparison group) will be conducted to determine significant mean differences using independent samples t-tests and chi-squared tests for continuous and categorical variables, respectively. Furthermore, analysis of covariance (ANCOVA) and repeated measures ANCOVA will be used to determine any significant between-group and within-group differences in the clinical measures and assessments after adjusting for covariates such as age, gender and baseline values. Finally, multivariate regression analyses using the backward stepwise method will be used to determine predictors of several clinical endpoints at 12-month follow-up. All analyses will be performed using R statistical software. Significance level is set as 0.05 and all statistical tests will be two-sided.

### 2.8. Ethical Considerations

The study was reviewed by the Western Sydney University Human Research Ethics Committee (REDI Reference: H12215). Written informed consent was obtained from the study participants.

## 3. Results and Discussion

Since the WellNet study is ongoing, the baseline findings on recruitment outcomes, baseline clinical and health assessment surveys along with provider level experiences at baseline are described. We have also discussed the strengths and limitations of the WellNet study along with potential determinants affecting study reproducibility.

### 3.1. Recruitment Outcomes

Almost 98% of the patients enrolled in the WellNet program were over 40 years of age and with 93% having one to three chronic conditions with one or more risk factors present, and 98% of participants had a HARP score in the >10 range (medium or greater risk). Thereby, it can be observed that the patient identification algorithm used was effective in identifying those patients who potentially met the program’s inclusion criteria (Table 3).

### 3.2. Sociodemographic Characteristics

The key demographic findings are presented in Table 4. The mean (SD) age of the study participants was 70.05 (11.6) years with nearly even gender distribution of males (49.7%) and females (50.3%). A majority of participants (68.7%) in the WellNet treatment group had private health insurance. There is higher than usual patients with private health insurance (PHI) because of higher socioeconomic status of North Sydney region and the PHI members targeted for intervention.

### 3.3. Chronic Disease Diagnosis and Clinical Indicators

The prevalence of chronic disease among the treatment and comparison group was similar. The most prevalent chronic diseases in descending order were: circulatory system disorders (69.8%), diabetes (47.4%), musculoskeletal disorders (43.5%), respiratory diseases (28.7%), mental illness (18.8%), and cancer (13.6%). Consistent with other Australian studies [66,67], the distribution of chronic diseases significantly differed across age groups as the number of chronic diseases generally increased with age (Figure 4). In terms of gender distribution, diabetes (54.8%) was significantly more prevalent amongst males than females (40.1%), whilst considerably more females had musculoskeletal disorders (54.3% vs. 31.3%), respiratory (34.1% vs. 23.8%), and mental illness (23.7% vs. 16.9%) than males.

Descriptive statistics of clinical measures for the treatment and comparison group are present in Table 5. Although treatment and comparison groups were effectively matched based on the type and number of chronic conditions, findings of the independent t-tests show significant differences in blood pressure, glycated haemoglobin (HbA1c), Low Density Lipoprotein Cholesterol (LDL-C), total cholesterol, and waist circumference (females only) between the treatment and comparison group. However, there were no statistically significant differences between treatment and comparison group for clinical measures like Body Mass Index (BMI), weight, waist circumference (males only), High Density Lipoprotein Cholesterol (HDL-C), estimated Glomerular Filtration Rate (e-GFR), and Albumin Creatinine Ratio (ACR) at baseline. In order to effectively manage baseline differences, ANCOVA will be used to determine any significant between-group differences in the clinical measures after adjusting for covariates such as age, gender and baseline values.

### 3.4. Self-Reported Health Assessments

General health and disease-specific risk assessments were recorded only among the WellNet treatment group (Table 6). As targeted, around 98% of participants had medium risk or greater HARP scores. The majority of study participants self-rated their health positively with mean EQ-5D-5L index score of 0.79 out of 1. Consistent with other studies [68,69], the mean EQ-5D-5L score significantly decreased with increased number of chronic diseases (*p* < 0.001). In terms of patient activation and self-management, only 21% reported having actively changed their health-related behaviour, with 22% beginning to take actions at baseline. However, 19% believed that action was not important, with a majority of participants (38%) reporting that the biggest difficulty in attempting to change behaviour was a lack of confidence or knowledge in managing and improving their condition.

### 3.5. Provider-Level Findings

#### 3.5.1. Provider-Level Baseline Characteristics

Of 66 eligible GPs, 49 (74%) responded to the self-reported survey. 62.5% of GPs were aged 55 and above and almost two-thirds (63.3%) had graduated in Australia. More than 63% reported more than 20 years of experience working in general practice.

#### 3.5.2. GPs Perspectives on Chronic Disease Management

At baseline, GPs were largely of the opinion (67% strongly agreed or agreed) that they had systematic approaches toward patients with chronic disease. Likewise, 70% of GPs reported using the skills and support of their practice nurses and primary care team. Around 90% stated that they routinely prepared chronic disease management plans for their patients and liaised with patients’ care teams. However, only 50% reported regularly reviewing and updating their disease management plans. How the latter number changes over the course of the twelve months of Phase 1 is of especial interest to the program.

#### 3.5.3. GPs Perspectives on WellNet Study

On the whole, GPs held positive perspectives toward the WellNet program. The majority of clinicians (>70%) agreed or strongly agreed that they were familiar with the concept of the program. GPs also reported to have excellent relationships with their primary care team (85% agreed or strongly agreed) as well as had confidence that clinical teams were providing highly coordinated care (77%).

Thirty-seven GPs identified and referred patients to the WellNet program and were confident that patients would benefit from the program. However, eleven stated that they had not. Of these, the most common reason was that the doctors in question did not know how to identify and refer patients to the program (n = 5). If these numbers have not changed after twelve months, greater clinician engagement may be necessary. One clinician stated that they did not believe the program had benefits to their patients, whilst one GP each stated that they had not identified any patients yet, had just started the program, or did not have any eligible patients.

#### 3.5.4. GPs Perspectives on PCMH Model

On the whole, it seems that clinicians were relatively aware of the PCMH concept (61% strongly agreed or agreed). These numbers should be expected to improve over twelve months. The majority of GPs were also optimistic that voluntary patient enrolment can enhance patient care (79%) and that they involved clinicians in the design of health service delivery (83%), while most also believed that they worked collaboratively in their practices and that clearly defined roles and responsibilities existed amongst staff.

It should also be noted that clinicians strongly valued patient feedback: nearly 40% strongly agreed, while 55% agreed. That said, only 71% agreed or strongly agreed that they had sufficient data for evaluating and improving their practices, and 81% believed that they were encouraged or supported to improve care quality and safety—indicating that there may be gaps in the collection or utilisation of patient outcome and experience data or insufficient structures in place that could support quality improvement.

### 3.6. Factors Affecting Reproducibility of Study Findings

Similar to other studies based on an originally developed program, reproducibility of study findings is constrained by the uniqueness of the data and also by several patient and provider-level determinants. Patients’ levels of education, social gradient, and health behaviour and practices would have different levels of impact towards improved health coaching and self-management practices leading subsequently to varying levels of management. For instance, patients with a higher level of education will better perceive the importance of self-management and would be more likely to adhere to health coaching provided by the GPs and care coordinators in the WellNet program. In terms of provider-level determinants, varying levels of experience and training among GPs and care coordinators will impact quality of care in terms of effectively delivering PCMH care.

### 3.7. Study Strengths

Despite general practices being the first point of health-related interactions among the predominant percentage of Australians, research based on GP data is disproportionately low [25]. To our knowledge, the WellNet program is the first study in Australia to generate the first real-world evidence on the feasibility and effectiveness of a comprehensive PCMH model with the use of primary care-based data. The study comprises a large, effectively targeted sample with wide range of GP-based data on chronic disease diagnosis as well as self-reported risk assessments. The use of general and disease-specific self-reported health assessments recorded among study participants helps understand patients’ quality of life, self-management practices, and severity of diseases in addition to routine clinical metrics. Furthermore, use of validated questionnaires renders better quality of data with improved credibility as well as higher comparability with other sample populations. In addition, the study also includes standardised clinical measurements that are collected by trained health care professionals as opposed to self-reported information, thereby eliminating any potential self-reporting bias.

### 3.8. Study Limitations

The WellNet program has some limitations in terms of the evaluation study design and data. The WellNet study was based on the case series study design for the treatment group only and did not include a control group. Therefore, analyses of self-reported assessments were limited to ‘within group’ comparisons rather than ‘between groups’. Although the treatment group was closely matched with the comparison group based on age, gender, type and number of chronic diseases, there were statistically significant differences observed across some clinical measures between the two groups at baseline. To minimize this, ANCOVA techniques will be used to control for baseline differences. In terms of data limitations, the reason why 52 patients declined from participation was not recorded. The impact of each component of PCMH contributing to primary and secondary outcomes was not explored. In addition, facilitators or barriers that contributed to early implementation experiences were also not recorded.

## 4. Conclusions

There has been increased advocacy to implement PCMH models for effective management at the primary care level worldwide. The WellNet study is one of the few studies in Australia to generate evidence on the feasibility of design, recruitment, and implementation of a comprehensive PCMH model. To date, the recruitment outcomes of WellNet study indicate that the patient algorithm was effective in identifying eligible patients for enrolment.

Lessons learnt from the baseline findings include understanding of the characteristics of the sample participants not only in terms of their chronic disease diagnosis, but also to assess their level of self-management behaviours, quality of life and other disease-specific surveys at baseline. As the PCMH model aims to provide a structured patient-tailored chronic disease management with focus on self-management behaviours, the baseline lifestyle and other health behaviours of the sample participants recorded at the start of the program enables GPs to better understand individual needs and provide better quality of care.

However, upon completion, we expect that the findings of WellNet study will inform patients on the benefits of the PCMH model in terms of improved clinical outcomes and quality of care, as well as reduced risk of hospitalisation.

## Figures and Tables

**Figure 1 ijerph-17-02164-f001:**
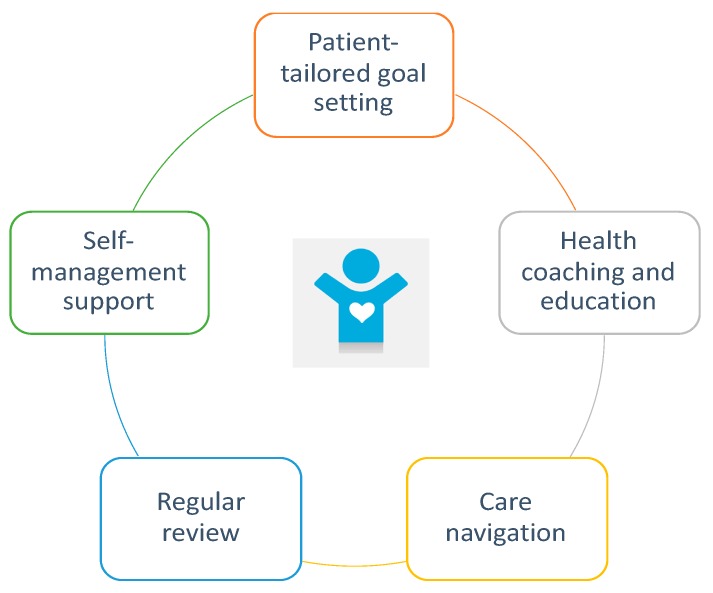
Patient-centred medical home (PCMH) components used in the WellNet model.

**Figure 2 ijerph-17-02164-f002:**
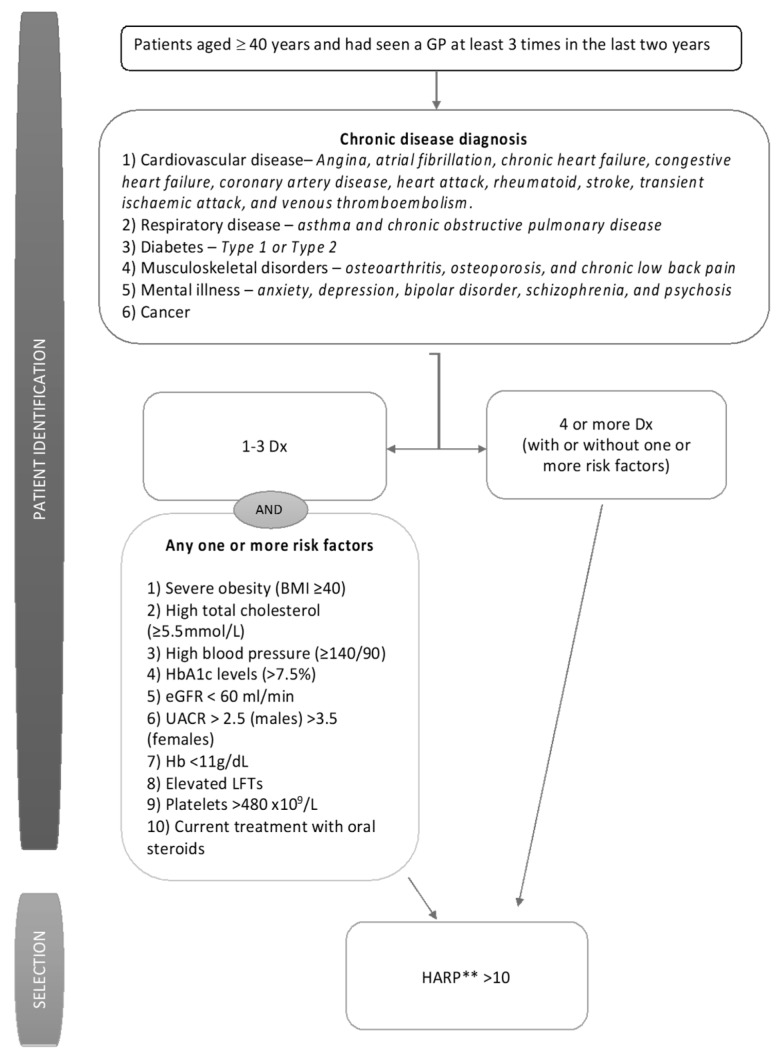
Eligibility criteria for the WellNet study. Dx–Diagnosis of chronic condition; BMI–Body Mass Index; HbA1c–Glycated Haemoglobin; eGFR–estimated Glomerular Filtration Rate; UACR–Urine Albumin Creatinine Ratio; LFT–Liver function test. ** Additionally, patients with low Hospital Admission Risk Profile (HARP) score (<10) but with at least one or more chronic diseases and one or more consistently elevated risk factors were also included in the study through general practitioner (GP) referrals.

**Figure 3 ijerph-17-02164-f003:**
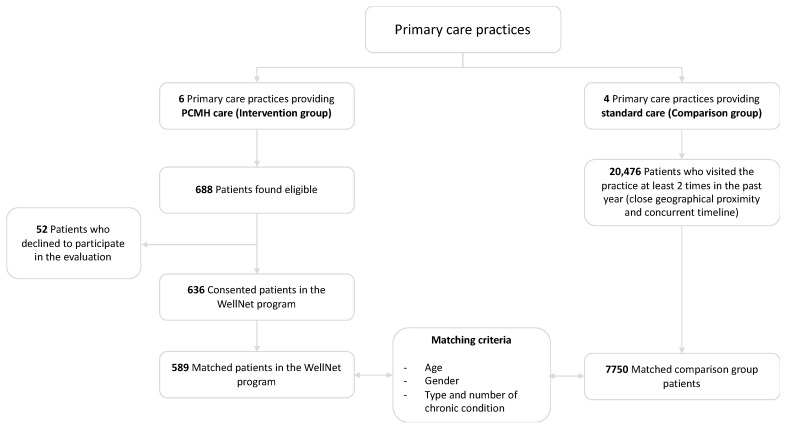
Flowchart of patient enrolment.

**Figure 4 ijerph-17-02164-f004:**
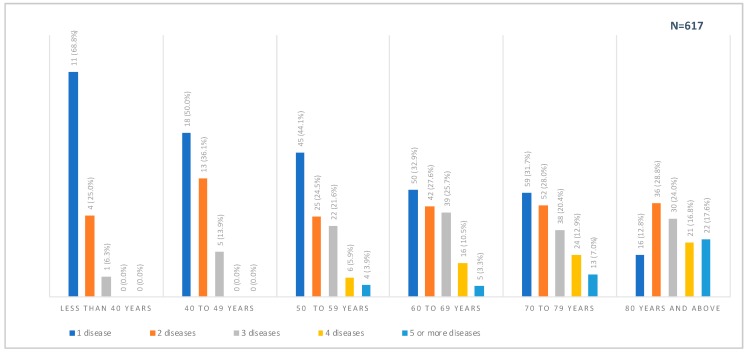
Distribution of chronic disease prevalence by age groups among WellNet patients. Chi-squared test between age group and number of chronic conditions returned a *p* < 0.001.

**Table 1 ijerph-17-02164-t001:** Matching iterations and outcomes.

Matching Iteration Number	Matching Variables	Treatment (Overall n = 636)	Comparison (overall n = 20478)	N (Matched Treatment: Comparison)	N (Unmatched Treatment: Comparison)
1	Age (continuous), gender, type of chronic disease (cardiovascular disease, respiratory disease, diabetes, musculoskeletal disease, mental illness, and cancer) and number of chronic diseases.	617 *	20478 *	589:7750	28:12728
2	Age (continuous), gender, type of chronic disease (cardiovascular disease, respiratory disease, diabetes, musculoskeletal disease, mental illness, and cancer), systolic, and total cholesterol: HDL-C ratio.	447 **	3899 **	228:604	219:3205
3	Age categories (breaks = 10 years), gender, (cardiovascular disease, respiratory disease, diabetes, musculoskeletal disease, mental illness, and cancer), systolic, and total cholesterol: HDL-C ratio.	447 **	3899 **	260:749	260:749
4	Age (continuous), gender, type of chronic disease (cardiovascular disease, respiratory disease, diabetes, musculoskeletal disease, mental illness, and cancer), systolic (breaks = 2 mmHg), and total cholesterol: HDL-C ratio.	447 **	3899 **	202:517	245:3292
5	Age categories (breaks = 10 years), gender, (cardiovascular disease, respiratory disease, diabetes, musculoskeletal disease, mental illness, and cancer), systolic (breaks = 2 mmHg), and total cholesterol: HDL-C ratio.	447 **	3899 **	242:735	205:3074

HDL-C–High Density Lipoprotein Cholesterol; * Of the 636 WellNet participants, 19 did not have a chronic disease and were not included in the matching analysis, resulting in 617 participants; ** Out of 636 WellNet participants, 19 did not have a chronic disease, 15 did not have a systolic reading and 166 did not have a TC: HDL reading, and they were automatically removed during matching. Similarly, out of 20478 comparison patients, 9176 did not have a chronic disease, 3169 did not have a systolic reading and 4234 did not have a TC: HDL reading, and they were automatically removed during matching.

**Table 2 ijerph-17-02164-t002:** Summary of data collected in the WellNet Study.

Type of Data	Time of Data Collection	Variables Measured
**Socio-demographic data**	Baseline	Age and gender
**Private health insurance membership**	Baseline	Private insurance status and name of the insurance provider
**Diagnosis of chronic condition**	Baseline	Diagnosis of arthritis, asthma, back pain, cancer, cardiovascular diseases (CVD), chronic obstructive pulmonary disease (COPD), diabetes, mental illness and kidney diseases.
**Clinical assessments**	Baseline 6-month 12-month	Height, weight, waist circumference, Body Mass Index (BMI), blood pressure (BP), blood sample - cholesterol [serum total cholesterol, high density lipoprotein cholesterol (HDL-C) and low-density lipoprotein cholesterol (LDL-C)], triglycerides, glycated haemoglobin (HbA1c), estimated glomerular filtration rate (eGFR), and urine albumin-creatinine ratio (UACR).
**Risk of hospital admission**	Baseline 12-month	Hospital Admission Risk Profile (HARP) score to estimate the risk of hospitalisation in the next 12 months
**Health utility data**	Baseline 12-month	Medication–drug name, dosage and frequency of use and number of prescriptions.
**Patient activation, engagement and readiness to change**	Baseline 12-month	Patient Activation Measure (PAM) scores to evaluate patient self-efficacy and self-management behaviours
**Patient self-reported health-related quality of life**	Baseline 12-month	EuroQol 5 Dimensions 5 Levels (EQ-5D-5L) to assess patient’s heath related quality of life
**Disease-specific risk assessments**	Baseline 12-month	Absolute Cardiovascular Disease Risk (CVDR), Australian Type 2 Diabetes Risk Tool (AusDrisk) scores, UK Prospective Diabetes Risk Engine (UKPDS), Chronic Obstructive Pulmonary Disease Assessment Test (CAT) scores, Kessler Psychological Distress scale (K10) scale, Depression Anxiety Stress Scale (DASS21), short versions of Knee Injury and Osteoarthritis Outcome Score (KOOS), Hip Disability and Osteoarthritis Outcome Score (HOOS).
**Patient experience survey**	12-month	Patient satisfaction survey on quality of care outcomes; patient-provider relationship, communication, and empowerment outcomes; and access to information technology outcomes.
**Feasibility outcomes**	12-month	Recruitment: number of patients’ who potentially met the program’s inclusion criteria; Retention rates: number of completed and dropouts; Treatment compliance: rates of adherence to overall protocol.

**Table 3 ijerph-17-02164-t003:** Feasibility outcomes.

Criteria	Success Rate
Age	
40 years and above	97.5%
Number of chronic diseases	
Number of patients with 1 to 3 chronic diseases	93.4%
Number of patients with 4 or more chronic diseases	6.6%
Risk factors	
Patients with 1 chronic disease (n = 227)	
1 risk factor	47.6%
>1 risk factor	31.7%
Patients with 2 chronic diseases (n = 237)	
1 risk factor	42.6%
>1 risk factor	27.4%
Patients with 3 chronic diseases (n = 115)	
1 risk factor	40%
>1 risk factor	35.7%
Patients with 4 or more chronic diseases (n = 38)	
1 risk factor	44.7%
>1 risk factor	26.3%
HARP score >10	97.7%

**Table 4 ijerph-17-02164-t004:** Sociodemographic characteristics and prevalence of chronic conditions in the treatment and comparison group.

Determinants	WellNet Treatment Group (N = 589)	WellNet Comparison Group (N = 7750)	*p*-Value
	N (%)	N (%)	
Age groups			0.798
40–44 years	7 (1.2)	141 (1.8)	
45–54 years	57 (9.7)	722 (9.3)	
55–64 years	118 (20.0)	1580 (20.4)	
65–74 years	188 (31.9)	2360 (30.4)	
75–84 years	156 (26.5)	2029 (26.2)	
≥85 years	63 (10.7)	918 (11.8)	
Mean (SD)	70.05 (11.59)	69.95 (11.86)	
Sex			0.936
Male	293 (49.7)	3842 (49.6)	
Female	296 (50.3)	3908 (50.4)	
Smoking status			
Ex–smoker	237 (42.1)	2274 (36.4)	<0.001
Non–smoker	280 (49.7)	3655 (58.6)	
Smoker	46 (8.2)	309 (5.0)	
Unknown	26 (4.4)	1512 (19.5)	
Drinking status			
Drinker	256 (43.5)	1465 (18.9)	<0.001
Non–drinker	176 (29.9)	147 (1.9)	
Unknown	156 (26.5)	6138 (79.2)	
Private insurance status			
Yes	404 (68.7)	NA	–
No	184 (31.3)		
Missing	1 (0.2)		
Prevalence of chronic conditions			1.000
Diseases of the circulatory system	411 (69.8)	5408 (69.8)	
Respiratory diseases	169 (28.7)	2224 (28.7)	
Musculoskeletal disorders	256 (43.5)	3368 (43.5)	
Diabetes	279 (47.4)	3671 (47.4)	
Mental illness	111 (18.8)	1461 (18.8)	
Cancer	80 (13.6)	1053 (13.6)	
Number of chronic conditions			1.000
1 disease	133 (22.6)	1750 (22.6)	
2 diseases	249 (42.3)	3276 (42.3)	
3 diseases	159 (27.0)	2092 (27.0)	
4 diseases	42 (7.1)	553 (7.1)	
≥5 diseases	6 (1.0)	79 (1.0)	

SD-Standard Deviation; NA—not available.

**Table 5 ijerph-17-02164-t005:** Clinical measures among treatment and comparison group collected at baseline.

Clinical Measures	WellNet Treatment Group (N = 589)	WellNet Comparison Group (N = 7750)	*p*-Value
	N (%)	N (%)	
Systolic blood pressure (mmHg), Mean (SD)	139.37 (19.30)	135.96 (16.51)	<0.001
Diastolic blood pressure (mmHg), Mean (SD)	79.27 (10.32)	76.02 (11.31)	<0.001
Body Mass Index, Mean (SD)	29.54 (6.36)	29.12 (5.63)	0.151
Weight (kg)–Mean (SD)	81.81 (20.72)	81.73 (19.70)	0.930
Waist Circumference* (Males) (cm), Mean (SD)	106.09 (15.63)	104.30 (14.30)	0.120
Waist Circumference* (Females) (cm), Mean (SD)	98.82 (14.62)	93.15 (12.47)	<0.001
HbA1c (%), Mean (SD)	7.16 (1.41)	6.71 (1.24)	<0.001
HDL-C (mmol/L), Mean (SD)	1.37 (0.40)	1.40 (0.42)	0.147
LDL-C (mmol/L), Mean (SD)	2.71 (1.08)	2.52 (0.98)	<0.001
Serum total cholesterol (mmol/L), Mean (SD)	4.81 (1.38)	4.60 (1.13)	<0.001
Estimated Glomerular Filtration Rate (mL/min/1.73m^2^)			
*>90 mL/min/1.73 m^2^*	109 (24.9)	1467 (24.8)	0.868
*60–89 mL/min/1.73 m^2^*	247 (56.5)	3436 (58.2)	
*45–59 mL/min/1.73 m^2^*	46 (10.5)	622 (10.5)	
*30–44 mL/min/1.73 m^2^*	26 (5.9)	274 (4.6)	
*<30 mL/min/1.73 m^2^*	9 (2.0)	106 (1.8)	
Missing	152 (25.8)	1845 (23.8)	
Albumin-Creatinine Ratio* (mg/mmol) (Males), Mean (SD)	15.52 (67.54)	13.08 (53.18)	0.653
Albumin-Creatinine Ratio* (mg/mmol) (Females), Mean (SD)	7.53 (22.31)	8.07 (32.12)	0.878

SD - Standard deviation; HbA1c–Glycated Haemoglobin; HDL-C - High Density Lipoprotein Cholesterol; LDL-C - Low Density Lipoprotein Cholesterol; p-value by Chi-square test (categorical) or independent samples t-test (continuous) between treatment and control group; *Variables with gender-specific cut-off points.

**Table 6 ijerph-17-02164-t006:** General and disease-specific assessments among WellNet treatment group only at baseline.

Patient Survey Questionnaires	WellNet Treatment Group (N = 635)
	n (%)
Hospital Admission risk profile (N = 628)	
Low risk (1–10)	14 (2.2)
Medium risk (11–23)	581 (92.5)
High risk (24–38)	33 (5.3)
Missing	7 (1.1)
Patient Activation Measure scores (N = 626)	
Not believing that activation is important (<47)	121 (19.3)
A lack of knowledge and confidence to take action (47.1–55.1)	236 (37.7)
Beginning to take action (55.2–67)	138 (22.0)
Taking action (>67.1)	131 (20.9)
Missing	9 (1.4)
Mean EQ-5D-5L score (overall)–Mean (SD)	0.79 (0.19)
EQ-5D-5L score percentage	
EQ-5D-5L mobility (N = 626)	
No problem	298 (47.6)
Slight problem	171 (27.3)
Moderate problem	108 (17.3)
Severe problem	47 (7.5)
Unable to walk	2 (0.3)
Missing	9 (1.4)
EQ-5D-5L self-care (N = 623)	
No problem	521 (83.6)
Slight problem	79 (12.7)
Moderate problem	19 (3.0)
Severe problem	4 (0.6)
Unable	0 (0.0)
Missing	12 (1.9)
EQ-5D-5L usual activities (N = 626)	
No problem	312 (49.8)
Slight problem	187 (29.9)
Moderate problem	98 (15.7)
Severe problem	26 (4.2)
Unable	3 (0.5)
Missing	9 (1.4)
EQ-5D-5L pain/discomfort (N = 627)	
No problem	156 (24.9)
Slight problem	246 (39.2)
Moderate problem	165 (26.3)
Severe problem	55 (8.8)
Extreme	5 (0.8)
Missing	8 (1.3)
EQ-5D-5L anxiety/depression (N = 629)	
No problem	321 (51.0)
Slight problem	190 (30.2)
Moderate problem	94 (14.9)
Severe problem	16 (2.5)
Extreme	8 (1.3)
Missing	6 (0.9)
DASS21 Scores (N = 331)	
DASS21–Depression scale	
Normal	230 (69.5)
Mild	26 (7.9)
Moderate	40 (12.1)
Severe	12 (3.6)
Extremely severe	23 (6.9)
DASS21–Anxiety scale	
Normal	211 (63.7)
Mild	32 (9.7)
Moderate	53 (16.0)
Severe	14 (4.2)
Extremely severe	21 (6.3)
DASS21–Stress scale	
Normal	260 (78.5)
Mild	29 (8.8)
Moderate	20 (6.0)
Severe	15 (4.5)
Extremely severe	7 (2.1)
K10 scores (N = 302)	
Low level of psychological distress	137 (45.4)
Moderate level of psychological distress	88 (29.1)
High level of psychological distress	49 (16.2)
Very high level of psychological distress	28 (9.3)
UKPDS risk profile (N = 140)	
Coronary Heart Disease risk - mean percentage (SD)	18.98 (14.11)
Fatal Coronary Heart Disease risk - mean percentage (SD)	14.94 (16.14)
Stroke risk - mean percentage (SD)	14.37 (15.10)
Fatal stroke risk - mean percentage (SD)	2.33 (2.60)
Missing	53 (27.5)
AusDRisk scores (N = 220)	
Low risk	5 (2.3)
Intermediate risk	34 (15.5)
High risk	181 (82.3)
Missing	137 (38.4)
Absolute cardiovascular risk (N = 370)	
Low risk	167 (45.1)
Moderate risk	84 (22.7)
High risk	119 (32.2)
Missing	48 (11.5)
COPD impact scores (N = 26)	
Normal	2 (7.7)
Low	4 (15.4)
Medium	12 (46.2)
High	3 (11.5)
Very high	5 (19.2)
Missing	38 (59.4)
HOOS score (N = 31)	
HOOS pain score - Mean (SD)	64.92 (23.37)
HOOS function score - Mean (SD)	68.95 (19.80)
HOOS symptoms score - Mean (SD)	66.15 (17.23)
Missing	168 (84.4)
KOOS score (N = 59)	
KOOS stiffness score - Mean (SD)	61.86 (28.37)
KOOS pain score - Mean (SD)	64.72 (22.01)
KOOS function score - Mean (SD)	63.13 (25.58)
KOOS symptoms score - Mean (SD)	63.09 (18.14)
Missing	140 (70.3)

SD—Standard Deviation; EQ-5D-5L—Euro-Qol Five dimensions and Five levels; DASS21—Depression Anxiety Stress Scale 21 questions version; K10—Kessler’s Psychological Distress scale; AusDRisk—Australian Type-2 Diabetes Risk score; COPD—Chronic Obstructive Pulmonary Disease; HOOS—Hip Disability and Osteoarthritis Outcome Score; KOOS—Knee Injury and Osteoarthritis Outcome Score.

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
