# Peer review of "Cohort Profile: Effectiveness of a 12-Month Patient-Centred Medical Home Model Versus Standard Care for Chronic Disease Management among Primary Care Patients in Sydney, Australia"

_ijerph, 2020, doi:10.3390/ijerph17062164_

Round 1
Reviewer 1 Report
This paper presents some background and research design material and some baseline data from a large quasi-experimental study of a PCMH project in Australia. PCMH and its application in Australia will be of interest to readers, but this paper provides a lot of baseline data but produces little useful data or theoretical information that others can use.
The fact that the study is about early implementation experiences and baseline data is not made clear in the abstract or in the early parts of the paper. The goal(s) and the rationale for this paper are not made clear. The paper touches on but does not explicate a number of interesting decisions in research design and early implementation, such as: choice of synthetic comparison group design, choices of matching procedure, socio-demographic/insurance status of the sample relative to the study communities, engagement of clinical sites, admissions of persons who did not meet all study criteria (presumably because clinicians saw these persons as good candidate for PCHM care for other reasons that are not explored. The study hints at bu does not address in detail the range early implementation experiences (that could be explored using the REACH or similar models for doing this).
The study reports extensively on the specific chronic conditions of subjects and the number of comorbid conditions and references "chronic disease type" in describing the matching process. It is unclear whether each pateint was categorized by a primary condition and number of other conditions or multiple chronic conditions and and number in the matching process.
The paper needs extensive editing and some rethinking to reflect its apparent goals, rather than presenting evaluation findings. For example, the analysis approach discussion in the methods section includes both research activities that were conducted so far and plans for future activities without clearly indicating what has been done. The reader has to guess that this paper is confined to early implementation experiences and baseline data. There is no attempt to justify how and why the presented baseline data is valuable....the study could examine, for example, how baseline values on the central planned outcome measures are related to the co-variates or how baseline data are linked to observed pre-admission utilization.
The PCMH program characteristics and provider adoption experiences are not described in detail. Did all providers asked to participate join? Did providers vary in rates of new patient enrollment and how might this be understood? How providers accessed and utilized the available data and clinical support resources what not explained. Reasons why about 50 folks decided not to participate are not explored.
From what is shared in this paper, the project sounds really fascinating in its complexity and how it draws on multiple prior PCMH experiences and new tools/other provider supports. And the quasi-experimental evaluation seems impressive. But this paper needs to much clearer about its goals and present, analyze and discuss the experiences to date in a way that can help other clinicians and researchers learn from its experiences.
Reviewer 2 Report
Well written manuscript. Very relevant as primary care clinics are moving forward with PCMH accreditation to provide patient-centered care and take advantage of value based payments.
Reviewer 3 Report
Brief summary:
This research sought to determine the relationship between Effectiveness of a 12-month patient centred medical home model of primary care versus standard care for chronic disease management of high risk patients in Sydney, Australia. Findings indicate that good interaction outcome between these determine factors, which might have an implication for theory and practice. Implementation of patient centred medical home model of primary care interaction outcome were associated with good chronic disease management of high risk patients.
Broad comments:
Although this study tackled an interesting topic and used a reasonably large sample, several methodological and statistical concerns preclude the acceptance of the manuscript for the publication according to this reviewer's perspective. In addition, the presentation of the study's rationale and reporting of the obtained findings requires a major reconstruction and rewriting, taking into account the clarity of statements and grammatical punctuality.
- From the title, abstract and other parts of the paper, the author's logic is not smooth, the key concepts are not explained clearly, and the relationship between related concepts is not explained.It only suggests that there are some differences, but the paper does not bring forward the conclusion of the related research.
- The use of core concepts in this paper is confusing and unclear.The concept and content of patient centred medical home model of primary care are not involved in this paper. The concept and content of health outcomes, shared-decision making, increased satisfaction through collaborative work practice, and improved safety and quality of care respectively. are not involved in this paper. The concepts of multimbidity、high risk patients and chronic disease are mixed in this paper.
Specific comments:
- P1 L5- high risk patients did not carry out relevant research in the paper, only mentioned briefly in P3L129.
- P1 L30-31 - Abstract - The study sample comprises 636 consented eligible participants in the intervention group who were matched with a comparison group of 7750 patients receiving usual GP care. The findings were not presented involved in the abstract.
- P1 L30-31 – P1Line L33 in the abstract is inconsistent with P6L203 in the text, Some place numbers are 623, some place numbers are 589. Inconsistent research core data in the paper
- P2 L56-65 -This part is not closely related to the research question. What is the purpose of this large section? Repeat the importance of research with P2L43-55.
- P2 L76-77 - It's not appropriate to replace the chronic diseases in the title of the paper with multimorbidity
- P2 L76-77 - Whether the research points of this paper are used in the research of the United States, whether there are differences and comparability between Australia and the United States.
- P3 L127-138 - Why was the cohort set up? What is the scientific nature of such a design.
- P3 L129- High risk patients is very abrupt here, which is not explained in the previous article.
- P7 L221- In this table, for example, where do the research indicators such as patient satisfaction survey come from? Why are these indicators comparative indicators not explained clearly.
- P7 L221- 2.4. Data collection: Who is in the cohort? Why do you choose such a research object? What are the representative articles with weak explanation.
- P7-L204-205-How often will the respondents be followed? It's only said that the tracking lasts for 12 months, but there's no specific explanation,the statement here is too general.
- P8 L241- 3. Results and discussion:The conclusion of T-tests along with Analysis of covariance method is not explained
- P14 L304-311 - This part of the content should be added, which is pale and powerless. There is no corresponding research conclusion in the paper to refine the innovation point.
- P14 L330-334 - In the conclusion, the concepts of job satisfaction, burnout rates, and patient provider relationship mentioned in this paper are not focused on discussion and data demonstration, but these results are not appropriate in the conclusion.
Round 2
Reviewer 1 Report
The paper is much clearer now and I appreciate the authors willingness to accommodate. I appreciate the addition of the physician survey.
I still think there is too much data provided.....across the tables there are many individual hypothesis tests---and no discussion of correcting p-values for multiple tests.....could you provide some of the data in an appendix? can you focus the text more clearly on the finding of similarity between the treatment and control group...you could offer some summary indicators of the % of comparisons that showed difference and highlight these in fewer tables..
Reviewer 3 Report
The author replied to all the 14 modification opinions put forward by the review and carefully revised them. The review found that they met the requirements and could be published!
